# CO_2_ Adsorption Performance and Kinetics of Ionic Liquid-Modified Calcined Magnesite

**DOI:** 10.3390/nano11102614

**Published:** 2021-10-05

**Authors:** Na Yang, Rong Xue, Guibo Huang, Yunqian Ma, Junya Wang

**Affiliations:** 1School of Environmental Science and Engineering, Qilu University of Technology (Shandong Academy of Sciences), Jinan 250353, China; yangna@qlu.edu.cn (N.Y.); xuerong1974@163.com (R.X.); chonglang_1982@163.com (G.H.); 2Faculty of Environmental Science and Engineering, Kunming University of Science and Technology, Kunming 650500, China

**Keywords:** ionic liquid, magnesite, CO_2_ capture, kinetics, nanomaterial

## Abstract

CO_2_ is a major contributor to global warming, and considerable efforts have been undertaken to capture and utilise it. Herein, a nanomaterial based on ionic liquid (IL)–modified calcined magnesites was investigated for CO_2_ capture. The synthesised nanomaterial (magnesite modified using [APMIM]Br) exhibited the best adsorption performance of 1.34 mmol/g at 30% IL loading amount, 50 °C, 0.4 MPa and 150 mL/min. In particular, the obtained nanomaterial could be regenerated at a low temperature of 90 °C for 3 h, and its CO_2_ adsorption capacity of 0.81 mmol/g was retained after eight cycles. FT-IR results showed that the imidazole ring and C–N group are directly related to CO_2_ adsorption capacity. Moreover, improving the conjugative effect of the imidazole ring enhanced the adsorption performance. Further, CO_2_ was adsorbed on the adsorbent surface and incomplete desorption decreased the BET surface area and CO_2_ adsorption capacity. Additionally, four models were selected to fit the adsorption kinetics. The results show that the adsorption mechanism fits the pseudo-first-order model well.

## 1. Introduction

CO_2_ emission reduction is crucial owing to the rapid increase in global warming. With the highest carbon emissions globally, China has pledged to stop the growth of CO_2_ emissions by 2030, followed by a gradual decline, thereby realising carbon neutrality by 2060 [1]. Carbon Capture, Utilisation and Storage (CCUS) technology is regarded as the only effective technology for achieving zero CO_2_ emissions in the aspect of the large-scale utilization of fossil energy [1]. Accelerating the research and development of CCUS technology and reducing costs and energy consumption are the main tasks at present [2]. CO_2_ capture is the core of CCUS technology. Generally, the CO_2_ adsorption capacity of liquid adsorbents is higher than solid adsorbents. However, high investment, corrosion of the equipment and high regeneration energy consumption are problems for liquid adsorbents applied in industry. Researchers across the globe have conducted studies on solid adsorbents, including layered double hydroxide-derived [3,4], CaO-based [5,6], MgO-based [7,8] and alkaline ceramic-based adsorbents [9,10] for pre-combustion CO_2_ capture, and carbon-based [11,12], zeolite-silica-based [13,14], metal-organic framework-based [15,16] and alkali metal carbonate-based adsorbents [17,18,19] for post-combustion CO_2_ capture. Furthermore, the adsorption process, such as two stage vacuum swing adsorption (VPSA), temperature swing adorption (TSA), temperature/vacuum swing adsorption (TVSA) [20] and sound-assisted fluidized bed reactor [21], has also been investigated for implementation in industrial applications.

In China, magnesite has nearly no value in utilization and is almost abandoned. If the abundant magnesite can be applied in CO_2_ capture, it will greatly reduce costs. In our previous work, MgO obtained by calcining low-cost magnesite was investigated at low temperature for post-combustion CO_2_ capture [22,23]. Although the CO_2_ adsorption capacity of MgO was sufficiently high, its regeneration temperature was extremely high, which is the biggest problem for its industrial application. Ionic liquids (ILs) can address such a problem and exhibit superior CO_2_ adsorption performance [24,25]; however, besides high cost, another drawback of pure ILs is their high viscosities causing strong gas diffusion resistances [26]. Therefore, for practical application, it will also be highly desirable to dispersedly immobilize ILs into a support in order to overcome the gas diffusion limitation. Therefore, we propose to modify calcined magnesite using ILs for CO_2_ capture. This approach can retain the benefits and solve the weakness of magnesite and ILs, respectively, designing an effective, cost efficient, easily achieved and low energy consumption adsorbent.

Herein, functionalised ILs with CO_2_ adsorption groups of –OH and –NH_3_, 1-aminoethyl-3-methylimidazolium bromide ([AEMIM]Br), 1-aminoethyl-3-methylimidazolium tetrafluoroborate ([AEMIM]BF_4_), 1-aminopropyl-3-methylimidazolium bromide ([APMIM]Br) and 1-aminopropyl-3-methylimidazolium tetrafluoroborate ([APMIM]BF_4_) were selected for investigation. Different adsorption conditions, such as the adsorption temperature, flow rate and relative pressure, were optimised to achieve parameters suitable for industry applications. Additionally, the thermal stability of the adsorbent was determined using thermogravimetry (TG) and differential scanning calorimetry (DSC) (TG–DSC). The structure and properties of the adsorbent were analysed using Scanning Electron Microscope (SEM), Fourier transform infrared (FT-IR) spectroscopy and the Brunauer–Emmett–Teller (BET) method. The adsorption kinetics were also calculated.

## 2. Materials and Methods

### 2.1. Reagents and Instruments

[AEMIM]Br, [AEMIM]BF_4_, [APMIM]Br and [APMIM]BF_4_, were of AR grade and obtained from Shanghai Chengjie Chemical Co., Ltd. (Shanghais, China). Ethanol (AR grade) was obtained from Tianjin Fuyu Fine Chemical Co., Ltd. (Tianjin, China). Magnesite was received from Laizhou Magnesium Mine, Shandong Province (Laizhou, China). N_2_ (99.999% purity) and 10% CO_2_ were supplied by Jinan Deyang Gas Co., Ltd. (Jinan, China). The gas mass flow controllers (S4932/MT) were purchased from Beijing Huibolong Instruments Co., Ltd. (Beijing, China). A rotary evaporator (RE-52A) was purchased from Shanghai Yarong Biochemical Instrument Factory (Shanghai, China). An electric heater (DF-101S) was obtained from Henan Yuhua Instrument Co., Ltd. (Gongyi, China). A muffle furnace (XMT806) was purchased from Longkou Xianke Instrument Co., Ltd., Shandong Province (Longkou, China). A portable infrared CO_2_ analyser (GXH-3010E), was supplied by Beijing HuaYun Instrument Co., Ltd. (Beijing, China). A counterbalance valve was purchased from Beijing Jiafa Instrument Co. Ltd. (Beijing, China). A powder compression machine (BJ-15) was purchased from Tianjin Bojun Technology Co., Ltd. (Tianjin, China). A magnetic stirrer, mortar and pestle, griddle and a fixed bed reactor as the adsorber were purchased from Jinan Bangen Instrument Co., Ltd. (Jinan, China).

### 2.2. Preparation of Adsorbents

The primary component of magnesite is MgCO_3_, and the appropriate calcination condition is 550 °C for 4 h [22]. Therefore, the magnesite was calcined in a muffle furnace for 4 h at 550 °C to obtain MgO. A certain amount of IL was added to 100 mL of ethanol and stirred continuously for 15 min using a magnetic stirrer. Based on the loading amount, the calcined magnesite was added to the aforementioned solution and stirred for 3 h to obtain a homogeneous mixture. To remove the solvent, the resulting solution was heated to remove the solvent in a rotary evaporator under vacuum. The obtained sample was pressed using a compression machine and then ground using a mortar. The resulting sample was sieved to screen solid particles with sizes 0.2–0.4 mm as the adsorbent for the test.

### 2.3. CO_2_ Adsorption-Regeneration Experiments

The CO_2_ adsorption apparatus is shown in Figure 1, including gas cylinders, relief valves, gas mass flow controllers, a fixed bed reactor and a CO_2_ analyser. Primarily, N_2_ was introduced to blow the gas analyser for 15 min at a flow rate of 100 mL/min. Thereafter, 10% CO_2_ simulated flue gas in the plant evacuated freely at a flow rate of 100 mL/min to stabilise the gas path. After attaining a steady state, 10% CO_2_ flowed to the adsorber with 3 g adsorbent by adjusting the three-way valve, while the CO_2_ analyser was turned on to perform the analysis. By changing IL loading amount, adsorption temperature, flow rate and relative pressure, the optimal adsorption conditions were determined. The temperature and flow rate were controlled using an electrical heating controller and a gas mass flow controller, respectively. The relative CO_2_ pressure was adjusted using a counterbalance valve.

The adsorbent regeneration experiment was performed by heating the saturated adsorbent under vacuum in a rotary evaporator for 3 h at 90 °C.

The CO_2_ adsorption capacity was calculated by the following equation:$${\mathrm{CO}}_{2}\;\mathrm{capacity}\;=\;\frac{A\times S\times M}{C\times m}\;\mathrm{mmol}/g$$
where *A* is the integral area by adsorption curves in min, *S* is the CO_2_ concentration, *M* is the CO_2_ flow rate in mL/min, *C* is constant 22.4 L/mol, *m* is the quality of adsorbent in g.

### 2.4. Characterisation of Adsorbents

SEM images were obtained using a ZEISS MERLIN Compact microscope (Carl Zeiss AG, Jena, Germany). The sample was coated with platinum. Particle size and size distribution analyses were performed on a dynamic light scattering device.

TG-DSC analysis was conducted using an SDT Q600 Universal V4.1D TA instrument (Waters Corp., Milford, MA, USA). The experimental material (3 mg) was purged with N_2_. The analysed temperature range was 25 °C to 750 °C and the heating rate was 10 °C/min. The thermal stability of the adsorbent was also tested.

FT-IR spectroscopic analysis was performed using a 5DXC IR spectrometer (Nicolet Instrument Corp., Madison, WI, USA). The wave scan ranged from 500–2000 cm^−1^, with a resolution of 2 cm^−1^. The sample testing was performed using KBr, and the sample was pressed into a pellet.

The specific surface area and porosity analysis were performed using a Tristar 3020 physisorption apparatus (MICROMERITICS Corp., Norcross, GA, USA). For this analysis, the BET model was selected for calculating the specific surface area, and the Barrett–Joyner–Halenda (BJH) method was used to analyse the pore size and distribution.

## 3. Results and Discussion

### 3.1. Properties of the Adsorbents

Figure 2 shows the infrared spectra of the magnesite modified using different types of ILs. The characteristic peak of the imidazole ring of the ILs appears at 1168 cm^−1^ and the C–N group peak is observed at 1334 cm^−^^1^, which indicates that the IL loading on the magnesite materials is successful. The peaks of magnesite modified using [AEMIM]BF_4_ and [APMIM]BF_4_ at 1083 cm^−^^1^ and 1126 cm^−^^1^ are caused by the B-F bond [25]. Moreover, the strength of the imidazole rings are in the order of [APMIM]Br > [AEMIM]Br > [AEMIM]BF_4_ > [APMIM]BF_4_.

The morphology features of the calcined magnesite and ILs-modified magnesite were determined using the obtained SEM image. Figure 3a,b show the SEM images of calcined magnesite the magnifications of 300 nm and 1 μm. The particles of the calcined magnesite (main component MgO [22]) show irregularities, and many nanoscale particles are clustered. Figure 3c–f are the SEM images of different ILs-modified magnesite and show the ILs loading on the magnesite successfully. All ILs evenly distribute on the magnesite. The particle sizes of the [AEMIM]Br and [APMIM]Br modified magnesite are uniform. [AEMIM]BF_4_ and [APMIM]BF_4_ modified magnesite are still clustered.

### 3.2. Influence of Different ILs

Four types of nanomaterials were fabricated by loading different types of ILs, ([AEMIM]Br, [APMIM]Br, [AEMIM]BF_4_ and [APMIM]BF_4_), at the loading amount of 10% on the calcined magnesite. The influence of the type of IL on the CO_2_ adsorption capacity was determined at the adsorption temperature of 50 °C and a gas flow rate of 100 mL/min. The adsorption curves of different modified magnesites are presented in Figure 4. This figure shows that the CO_2_ adsorption performance of the bromide-based magnesite is better than that of the tetrafluoroborate-based magnesite, with an effective adsorption time of >10 min. The CO_2_ effective adsorption times of the magnesite modified using [AEMIM]BF_4_ and [APMIM]BF_4_ are approximately 8 and 7 min, respectively. The CO_2_ adsorption capacity (Table 1) of the [AEMIM]Br-modified magnesite is 0.29 mmol/g, lower than that achieved by the [APMIM]Br-modified magnesite, i.e., 0.34 mmol/g. These results may be due to the fact that the tetrafluoroborate-based magnesite is gathered, and the particle size of bromide-based magnesite is uniform, which can be seen in the SEM images. Thus, the magnesite modified using [APMIM]Br was selected as the adsorbent in the subsequent experiments.

### 3.3. Influence of IL Loading Amount

The loading amount is directly related to the cost of ILs and CO_2_ adsorption performance. Therefore, the IL loading amount was investigated using the [APMIM]Br-modified magnesite as the adsorbent at the adsorption temperature of 50 °C and a gas flow rate of 100 mL/min. The results are presented in Figure 5 and Table 1. The CO_2_ capacity of the calcined magnesite without IL modification is only 0.20 mmol/g. Introducing ILs can improve the CO_2_ capacity of the calcined magnesite. With the loading amount increasing, the CO_2_ adsorption performance of the modified magnesite also improves. The effective adsorption time with 20% and 30% loading amounts is about 18 min, longer than that achieved with a 10% loading amount. The CO_2_ adsorption capacity of the modified calcined magnesite with 20% and 30% loading amounts can reach 0.76 and 0.95 mmol/g, respectively. Thus, ILs were important in capturing CO_2_. As the IL is too expensive, an extremely high loading amount implies a high cost, which is industrially inappropriate. Moreover, an excess IL amount might result in agglomeration, inhibiting CO_2_ capture. The magnesite modified using [APMIM]Br with a 30% loading amount was selected for subsequent experiments.

### 3.4. Influence of Adsorption Temperature

After desulphurisation, the temperature of the industrial flow gas can decrease from 100 °C to 50 °C; hence, the influence of temperature on the CO_2_ adsorption performance was investigated at 30, 50, 70 and 90 °C using the [APMIM]Br-modified magnesite with the loading amount of 30% and a gas flow rate of 100 mL/min. The results are presented in Figure 6 and Table 1, showing that the CO_2_ capacity is only 0.72 mmol/g at 30 °C and culminates to 0.95 mmol/g at 50 °C. As the temperature increases, the CO_2_ adsorption capacity gradually decreases, i.e., 0.81 mmol/g at 70 °C and 0.78 mmol/g at 90 °C. The influence of the temperature on CO_2_ capture is complex. In the temperature range of 30 °C to 50 °C, increasing temperature can improve the chemical adsorption on modified magnesite. However, when the temperature increases continuously, the CO_2_ desorption happens, leading to the decline of the adsorption capacity. The results also imply that a low temperature can satisfactorily modify the magnesite regeneration.

### 3.5. Influence of Gas Flow Rate

The adsorption efficiency should also be improved to realize industrial applications. Therefore, the CO_2_ adsorption performance of the magnesite modified by [APMIM]Br was studied at 50 °C and gas flow rates of 100, 150 and 200 mL/min. As shown in Figure 7 and Table 1, with the flow rate increasing, the effective adsorption time decreases and the CO_2_ adsorption capacity gradually decreases. By comprehensively considering the CO_2_ adsorption capacity and exhaust gas-pressing quantity, the gas flow rate of 150 mL/min was deemed to be appropriate for subsequent experiments, with an adsorption capacity of 0.85 mmol/g.

### 3.6. Influence of Relative Pressure

Our previous work reported that pressure can improve the CO_2_ adsorption capacity of calcined magnesite [22]. The CO_2_ adsorption performance of the magnesite modified by [APMIM]Br was also studied at 50 °C, a gas flow rate of 150 mL/min and relative pressures of 0, 0.2 and 0.4 MPa. The adsorption curves are presented in Figure 8, indicating that the effective adsorption time extends slightly as the relative pressure is increased, i.e., nearly 20 min at 0.4 MPa. When the relative pressure is increased from 0 to 0.4 MPa, the CO_2_ adsorption capacity increases from 0.85 to 1.34 mmol/g (Table 1), proving that the relative pressure also has an obvious influence on CO_2_ capacity.

### 3.7. Cyclic Regenerative Performance

Except for CO_2_ adsorption capacity, the regeneration of the adsorbent is also a key factor affecting industrial applications. The adsorption was performed using the magnesite modified by [APMIM]Br with 30% loading amount at 50 °C, 0.4 MPa and 150 mL/min, and desorption was performed at 90 °C for 3 h under vacuum in a rotary evaporator. Figure 9 shows that the CO_2_ adsorption capacity decreases from 1.34 to 0.95 mmol/g for the first cycle, showing a slightly downward trend from the second to the fifth cycle, after which it remains steady until the eighth cycle.

Compared with calcined magnesite and other MgO-based adsorbent in our previous work [22], the CO_2_ adsorption capacity of the [APMIM]Br-modified magnesite is 1.34 mmol/g, slightly lower than the calcined magnesite in the presence of water, i.e., 1.82 mmol/g. However, the advantage of the calcined magnesite modified by [APMIM]Br is that the regeneration temperature is decreased from 550 °C to 90 °C [22]. Both the adsorption temperature 50 °C and regeneration temperature 90 °C are in the range of flue gas temperature after desulphurisation (about 50~100 °C), so the flue gas energy can be used directly for CO_2_ capture and regeneration, reducing the consumption. The CO_2_ adsorption performance of other ILs-based adsorbents are shown in Table 2 [26,27,28,29,30,31,32]. By comparison, the 1.34 mmol/g CO_2_ capacity we achieved is lower than 5.53 mmol/g by TEOS-APTES-EMIM(Tf_2_N) and 4.49 mmol/g by [bmim][Ac]. However, the performance of [APMIM]Br-modified magnesite has little difference from the other ILs-based adsorbents listed in Table 2 in CO_2_ capacity and regeneration conditions. [APMIM]Br-modified magnesite retains the benefits of ILs-based adsorbent and the material used is non-pollution. Furthermore, the low ILs consumption, the abundant magnesite and the simple production process reduce the cost greatly for industry application. MOFs and activated carbon have been reported to achieve high CO_2_ adsorption capacity. However, the cost of production and raw materials are the significant challenging factors for industry application [33]. Activated carbons or porous carbons for CO_2_ capture need chemical activation (chemical agents and temperatures from 400 °C to 900 °C), physical activation (temperatures about 700 °C to 1200 °C), or others. Both activation process and regeneration require much energy, and the production of activated carbons also incurs costs [34]. Therefore, the [APMIM]Br-modified magnesite is available and could potentially be adopted on an industrial scale.

### 3.8. Characterisation

The TG-DSC curve of pure magnesite has been reported in our previous work [22], showing that the decomposition of magnesite to MgO happens from 400 °C to 800 °C. The TG-DSC curve of the [APMIM]Br-modified magnesite is shown in Figure 10. As the temperature is increased from 30 °C to 100 °C, the mass loss rate is approximately 7.3%, which can be attributed to the desorption of micromolecules or impurities, including H_2_O, CO_2_ and others adsorbed by the sample in air and residual solvents from the preparation process. A peak is observed at 200–450 °C, corresponding to the mass loss rate of 28.5%, accompanied by a DSC endothermic peak, which is attributed to the decomposition of the [APMIM]Br-modified magnesite. Therefore, at 30–100 °C, the [APMIM]Br-modified magnesite exhibits excellent thermostability.

Figure 11 shows the infrared spectra of the magnesite modified by [APMIM]Br with different loading amounts and the 30% loading amount of the modified magnesite after adsorption. The strength of both the imidazole ring characteristic and C–N group peaks at 1168 and 1334 cm^−1^, respectively, is in the order of the loading amounts of 30% > 20% > 10%. Moreover, the strength magnitudes of the peaks after CO_2_ adsorption decrease. The CO_2_ adsorption capacity determined experimentally of the different ILs and loading amounts are consistent with the strengthening of the imidazole ring and C–N groups (Figure 2 and Figure 11), indicating that improving the conjugative effect of the imidazole ring can enhance the adsorption performance.

The BET results are presented in Table 3. The N_2_ adsorption-desorption isotherm and the pore size distributions of the magnesite modified by [APMIM]Br before adsorption and after the first cycle are presented in Figure 12. Before adsorption, the BET surface area is 128.5 m^2^/g, the pore volume is 0.2 cm^3^/g and the pore size is 62.5 Å. After the first cycle, the BET surface area is 105.6 m^2^/g, the pore volume is 0.13 cm^3^/g and the pore size is 51.0 Å. The results verify that the modified magnesite used in the experiments shows a good pore structure, which plays an essential role in CO_2_ adsorption. The reduction in the BET surface area, pore volume and pore size after the first cycle may be attributed to CO_2_ adsorption on the modified magnesite surface and incomplete desorption, thus decreasing the CO_2_ adsorption capacity. By comparison with the BET surface area of the pure calcined magnesite, 118.6 m^2^/g [22], it indicates that the 30% IL loading amount has little influence on the pore structure of the calcined magnesite. According to the International Union of Pure and Applied Chemistry classification, the two samples belong to the type IV isotherm with an H3 hysteresis loop, coinciding with mesoporous materials (pore size 2–50 nm).

### 3.9. Adsorption Kinetics

To study the adsorption kinetics of CO_2_ on the magnesite modified by [APMIM]Br, the pseudo-first-order, pseudo-second-order, intraparticle diffusion and Bangham models were used to fit the adsorption curves [35,36].

The pseudo-first-order model is expressed as
*q_t_* = *q_e_*[1 − *exp*(−*k*_1_*t*)](1)
where *q_t_* is the CO_2_ adsorption capacity at *t* min in mmol/g, *q_e_* is the CO_2_ adsorption capacity at the adsorption equilibrium in mmol/g and *k*_1_ is the adsorption reaction rate in min^−1^.

The pseudo-second-order model is expressed as
*q_t_* = *k*_2_ × *q_e_*^2^*t*/(1 + *k*_2_*q_e_t*)(2)
where *k*_2_ is the adsorption reaction rate in g/mmol·min.

The intraparticle diffusion model is expressed as
*q_t_* = *k_i_**t*^0.5^ + *C*(3)
where *k_i_* is the intragranular diffusion constant in mmol·g^−1^ min^−0.5^ and *C* is a constant in mmol/g.

The Bangham model is expressed as
*loglog* [*q_e_*/(*q_e_ − q_t_*)] = *log* (*k_b_*/2.303) + *n*·*log t*(4)
and *q_t_* can be expressed as
*q_t_* = *q_e_*[1 − *exp*(−*k_b_t^n^*)](5)
where *k_b_* is the Bangham constant in min^−n^ and n is a constant.

The pseudo-first-order and pseudo-second-order indicate that the reaction rate is related to one or two reactants. The intraparticle diffusion model indicates that the overall adsorption process is controlled by several steps, film or external diffusion, pore diffusion, surface diffusion and adsorption on the pore surface, a combination of more than one step etc. The Bangham model indicates that the pore-diffusion is the only rate-controlling step [36,37].

The experimental dates and the non-linear fitting curves of the four adsorption kinetic models are shown in Figure 13. The kinetic parameters are listed in Table 4. The kinetic parameter R^2^ value of the pseudo-first-order and Bangham models is 0.999. The concluded *q_e_* values by pseudo-first-order 1.302 mmol/g and Bangham models 1.302 mmol/g are closed to the experimental date. The pseudo-second-order overestimates CO_2_ adsorption capacity (*q_e_*) (Table 4) in the first few minutes (about <8 min) and underestimates in the following minutes (about 8 min~32 min), with *R*^2^ value 0.988. The pseudo-first-order and Bangham models fit the experimental curves well. When the parameter *n* of Bangham is 1, the Bangham model is entirely the same as the pseudo-first-order model. Therefore, the adsorption mechanism fits the pseudo-first-order model excellently and the process should accord with the characteristics of the pseudo-first-order model. It can be concluded that the CO_2_ adsorption process involves reversible adsorbent surface interactions, and the rate is related to CO_2_ concentration directly.

## 4. Conclusions

Herein, an effective and easily achieved nanomaterial was prepared using calcined magnesite modified with [APMIM]Br, which had the benefits of cost efficiency, low energy consumption and non-pollution. By comparison, the particle sizes of [AEMIM]Br and [APMIM]Br modified magnesite were uniform and had higher CO_2_ capacity. [AEMIM]BF_4_ and [APMIM]BF_4_ modified magnesite were clustered and had lower CO_2_ capacity. Besides, the conjugative effect of the imidazole ring could affect the adsorption performance. By experiments, the optimal adsorption conditions of 30% IL loading amount, 50 °C, 0.4 MPa and 150 mL/min gas flow rate was achieved with a CO_2_ adsorption capacity of 1.34 mmol/g. The regeneration temperature was reduced to 90 °C, saving considerable amounts of energy, which is beneficial for industrial applications. The adsorbent showed good thermostability in the experimental temperature range. However, the cyclic stability needs to be further improved. The investigation of the adsorption kinetics indicated that the adsorption mechanism fits the pseudo-first-order model well.

## Figures and Tables

**Figure 1 nanomaterials-11-02614-f001:**
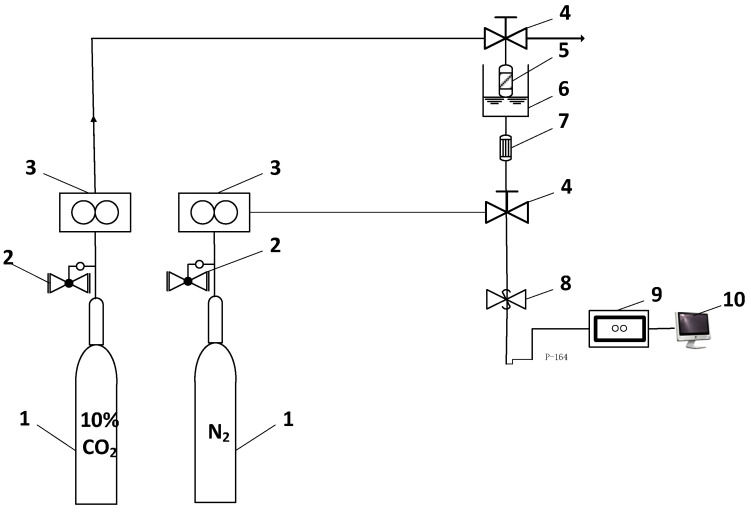
Apparatus for CO_2_ adsorption. (1) Gas cylinder, (2) relief valve, (3) gas mass flow controllers, (4) three-way valve, (5) fixed bed reactor, (6) electrical heater, (7) filter, (8) counterbalance valve, (9) infrared gas analyser, (10) data collector.

**Figure 2 nanomaterials-11-02614-f002:**
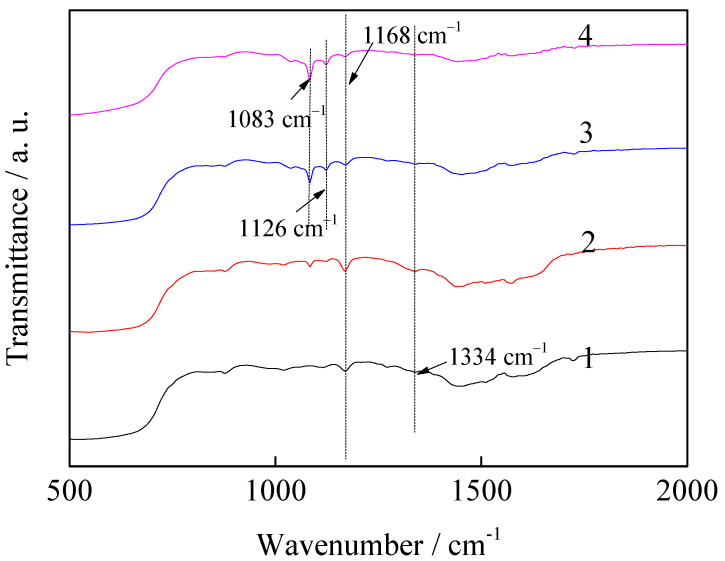
Infrared spectra of magnesite modified using different ILs ((1) [AEMIM]Br, (2) [APMIM]Br, (3) [AEMIM]BF_4_ and (4) [APMIM]BF_4_).

**Figure 3 nanomaterials-11-02614-f003:**
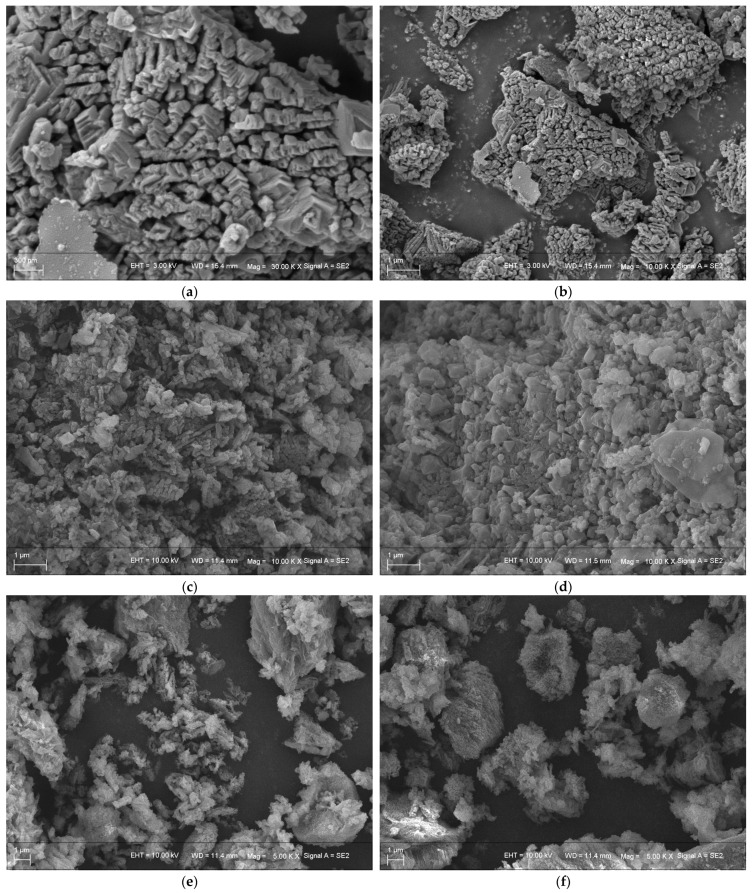
SEM images of calcined magnesite and ILs-modified magnesite with different magnifications: ((**a**) calcined magnesite (300 nm); (**b**) calcined magnesite (1 μm); (**c**) [AEMIM]Br-modified magnesite (1 μm); (**d**) [APMIM]Br-modified magnesite (1 μm); (**e**) [AEMIM]BF_4_ -modified magnesite (1 μm) and (**f**) [APMIM]BF_4_-modified magnesite (1 μm)).

**Figure 4 nanomaterials-11-02614-f004:**
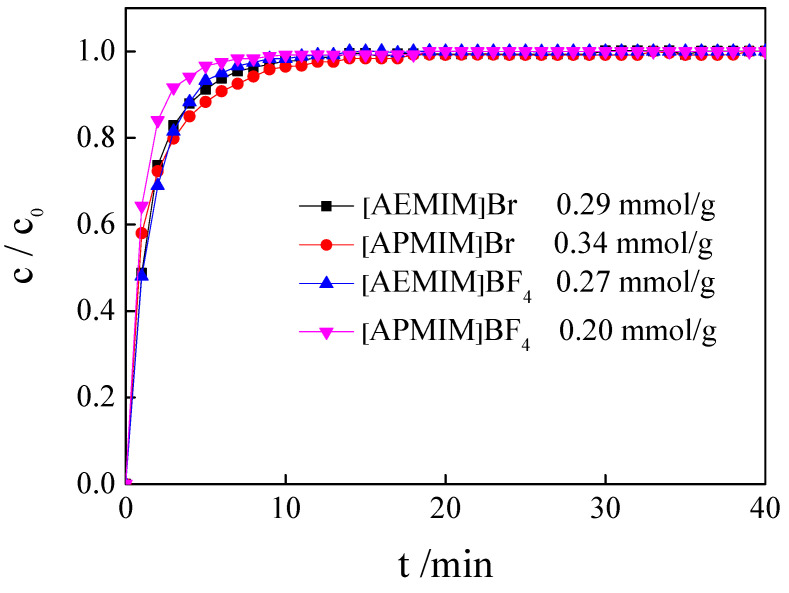
CO_2_ adsorption curves and CO_2_ capacity of different modified magnesites (3 g adsorbent, 10% loading amount, 100 mL/min flow rate, 50 °C and 0 MPa; c–CO_2_ monitoring concentration, c_0_–CO_2_ initial concentration, the same in the following figures).

**Figure 5 nanomaterials-11-02614-f005:**
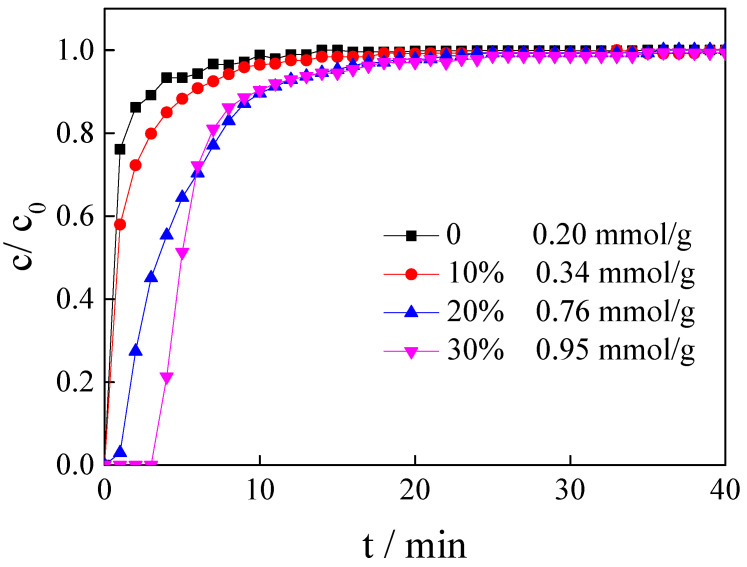
The CO_2_ adsorption curves and CO_2_ capacity of magnesite modified by [APMIM]Br with different loading amounts (3 g, 100 mL/min flow rate, 50 °C and 0 MPa).

**Figure 6 nanomaterials-11-02614-f006:**
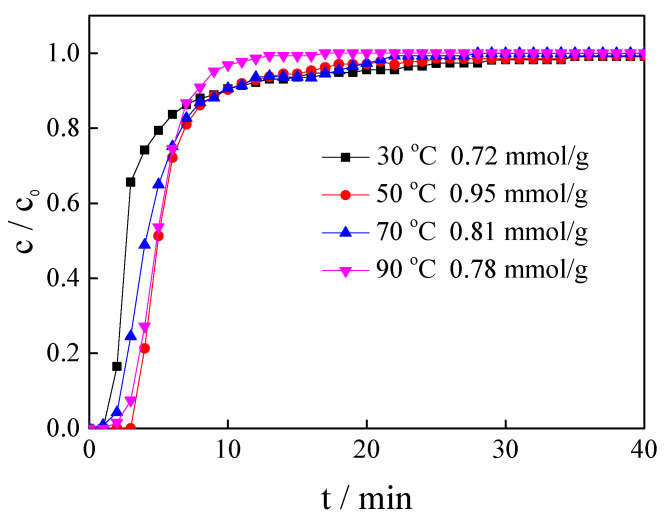
Adsorption curves and CO_2_ capacity of magnesite modified by [APMIM]Br at different temperatures (3 g, 30% loading amount, 100 mL/min flow rate and 0 MPa).

**Figure 7 nanomaterials-11-02614-f007:**
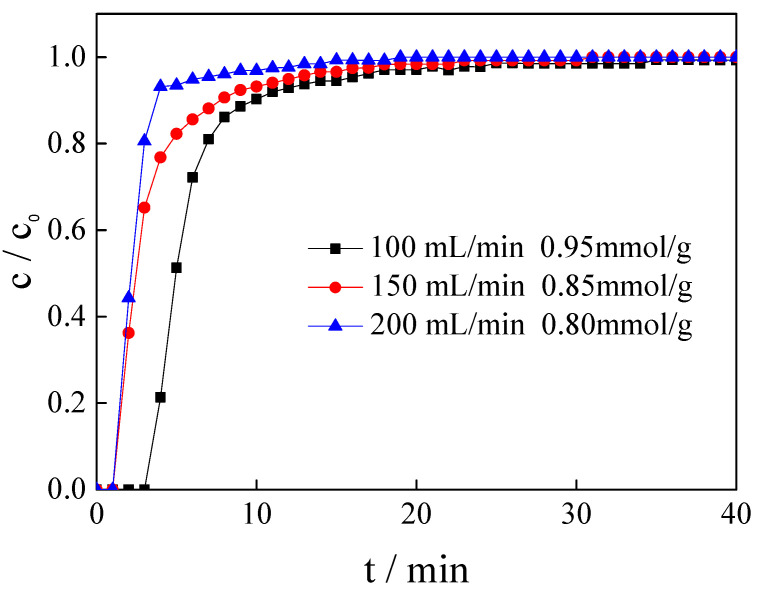
Adsorption curves and CO_2_ capacity of magnesite modified by [APMIM]Br under different flow rates (3 g, 10% loading amount, 50 °C and 0 MPa).

**Figure 8 nanomaterials-11-02614-f008:**
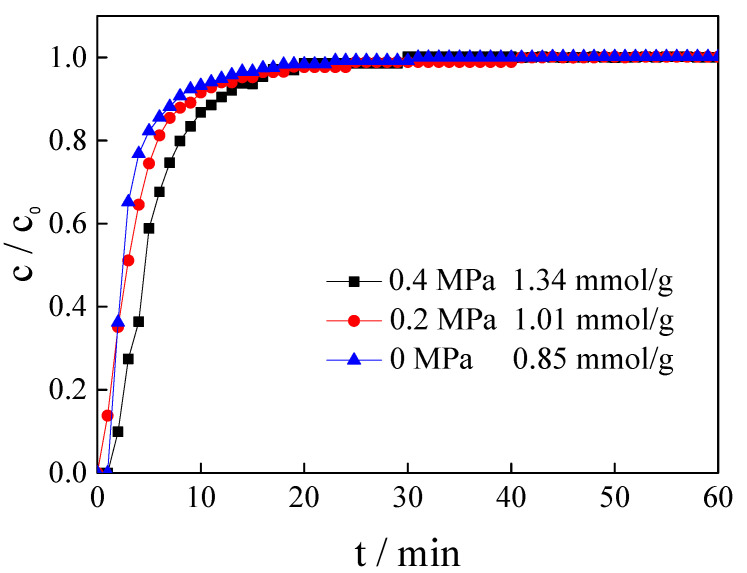
Adsorption curves and CO_2_ capacity of magnesite modified by [APMIM]Br under different pressures (3 g, 10% loading amount, 50 °C and 150 mL/min).

**Figure 9 nanomaterials-11-02614-f009:**
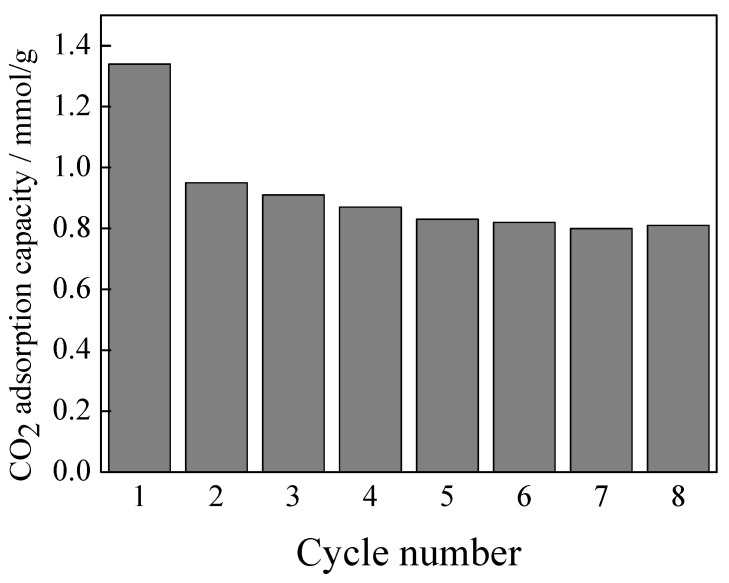
CO_2_ adsorption capacity of [APMIM]Br-modified magnesite for eight cycles.

**Figure 10 nanomaterials-11-02614-f010:**
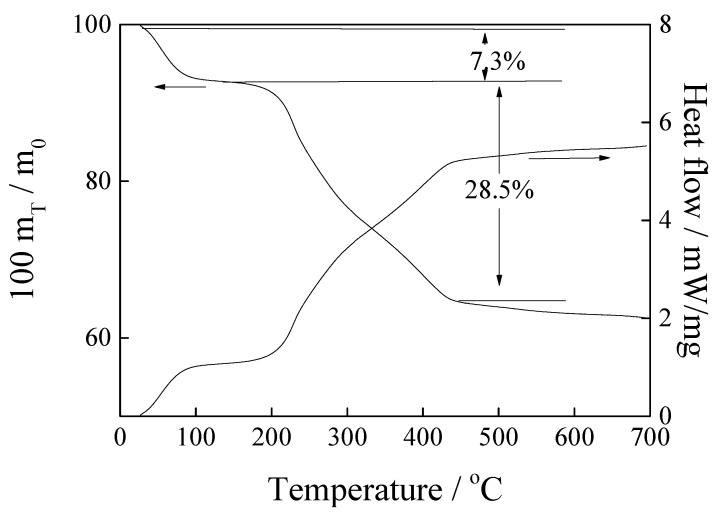
The TG-DSC curve of the magnesite modified by [APMIM]Br.

**Figure 11 nanomaterials-11-02614-f011:**
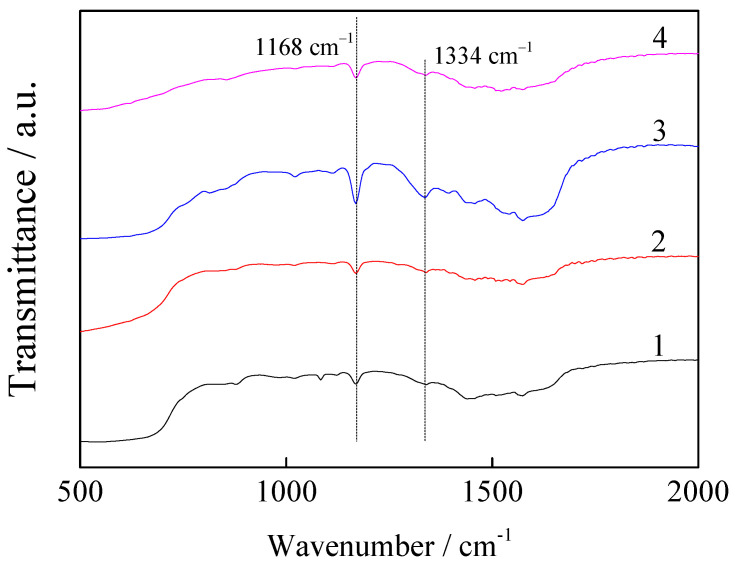
Infrared spectra of magnesite modified by [APMIM]Br with different loading amounts and 30% loading amount of the modified magnesite after adsorption ((1) 10% loading amount; (2) 20% loading amount; (3) 30% loading amount and (4) after adsorption).

**Figure 12 nanomaterials-11-02614-f012:**
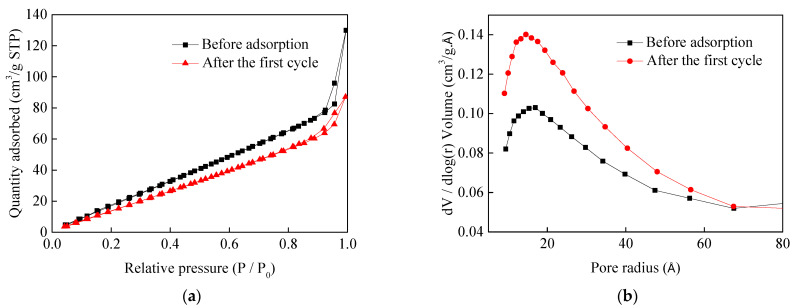
N_2_ adsorption-desorption isotherms (**a**) and pore size distributions (**b**) of magnesite modified by [APMIM]Br before and after the first cycle.

**Figure 13 nanomaterials-11-02614-f013:**
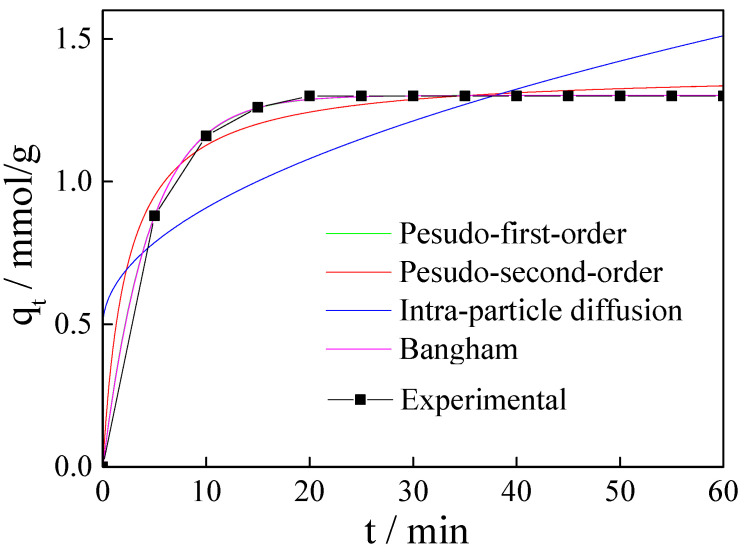
Adsorption kinetics of CO_2_ on magnesite modified by [APMIM]Br under the conditions of 50 °C, 0.4 MPa, gas flow rate 150 mL/min.

**Table 1 nanomaterials-11-02614-t001:** CO_2_ adsorption capacity of magnesite modified under different conditions (10% CO_2_).

Ionic Liquid	Loading Amount	Adsorption Temperature (°C)	Flow Rate (mL/min)	Adsorption Pressure (MPa)	CO_2_ Adsorption Capacity (mmol/g)
[AEMIM]Br	10%	50	100	0	0.29
[APMIM]Br	10%	50	100	0	0.34
[AEMIM]BF_4_	10%	50	100	0	0.27
[APMIM]BF_4_	10%	50	100	0	0.20
[APMIM]Br	0	50	100	0	0.20
[APMIM]Br	20%	50	100	0	0.76
[APMIM]Br	30%	50	100	0	0.95
[APMIM]Br	30%	30	100	0	0.72
[APMIM]Br	30%	70	100	0	0.81
[APMIM]Br	30%	90	100	0	0.78
[APMIM]Br	30%	50	150	0	0.85
[APMIM]Br	30%	50	200	0	0.80
[APMIM]Br	30%	50	150	0.2	1.01
[APMIM]Br	30%	50	150	0.4	1.34

**Table 2 nanomaterials-11-02614-t002:** Comparison of CO_2_ capacity and regeneration condition of ILs-based adsorbent.

Sample	Adsorption Temperature (°C)	CO_2_ Adsorption Capacity (mmol/g)	Regeneration Temperature (°C)	Reference
TEOS ^a^-APTES ^b^-EMIM(Tf_2_N)	-	5.53	-	[27]
[bmim][Ac] ^c^	30	4.49	-	[28]
[AEMIM][Lys] ^d^-immobilized on PMMA ^e^	30	1.50	100	[29]
PSF ^f^-[bmim][NTf_2_]-Fe_2_O_3_	45	1.30	70	[30]
[DMAPAH][EOAc]	30	2.44	30	[31]
[TBMP][MeSO_4_] ^g^	40	1.13	-	[32]
[EMIM][Lys]/PMMA	30	1.20	100	[26]
[APMIM]Br-modified magnesite	50	1.34	90	This work

^a^ TEOS represents tetraethyl orthosilicate. ^b^ APTES represents (3-Aminopropyl)triethoxysilane. ^c^ [Ac] represents acetate. ^d^ [Lys] represents L-lysine. ^e^ PMMA represents Poly(methyl methacrylate). ^f^ PSF represents Polysulfone. ^g^ [TBMP][MeSO_4_] represents tributylmethylphosphonium methylsulfate.

**Table 3 nanomaterials-11-02614-t003:** Porous structure parameters determined from N_2_ adsorption-desorption isotherms.

Sample	BET Surface Area (m^2^/g)	Pore Volume (cm^3^/g)	Pore Size (Å)
Magnesite modified by [APMIM]Br	128.5	0.2	62.5
Magnesite modified by [APMIM]Br after the first cycle	105.6	0.13	51.0

**Table 4 nanomaterials-11-02614-t004:** Kinetics parameters for magnesite modified by [APMIM]Br under the conditions of 50 °C, 0.4 MPa and gas flow rate 150 mL/min.

Models	Parameter	Value
Pseudo-first-order	*R* ^2^	0.999
	*k* _1_	0.224
	*q_e_*	1.302
Pseudo-second-order	*R* ^2^	0.988
	*k* _2_	0.312
	*q_e_*	1.386
Intraparticle diffusion	*R* ^2^	0.624
	*k_i_*	0.131
	*C*	0.491
Bangham	*R* ^2^	0.999
	*k_b_*	0.221
	*q_e_*	1.302
	*n*	1.010

## Data Availability

Not applicable.

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
