# Peer review of "CO_2_ Adsorption Performance and Kinetics of Ionic Liquid-Modified Calcined Magnesite"

_nanomaterials, 2021, doi:10.3390/nano11102614_

Round 1

Reviewer 1 Report

The authors did not answer all the referees' questions and did not highlight their changes, which made it difficult to track the changes. I do not consider this article for publication.

Some fundamental problems listed above remain at work:

  • The paper don’t bring great novelty (as new discussion or sorption mechanism, new material or good sorption capacity).
  • The use of the material for CO2 capture is not really viable, since it uses lot of energy for calcination and don’t show good recyclability.
  • The increase in the amount of IL generates an increase in the sorption capacity, which may indicate that the ILs pure or mixed between IL and any residue (not calcined) are capable of capturing more CO2 than the material.
  • The calcination part to produce the material consumes a lot of energy, and the authors have not demonstrated the advantage of this.
  • Blank experiments are missing.
  • "torus imidazole" is an unusual expression;
  • How can the relative pressure be 0 MPa?
  • All graphs (adsorption curves) are similar, and only show the adsorption rate, which does not represent the most relevant information for the article. It should be more interesting to move part of the graph to supporting information and create figures/graphs comparing the sorption capacity of each material.
  • - I am still not convinced with the kinetic experiment, the fit is similar in many models, this part should be further explored to prove its conclusion.

Author Response

Response to Reviewer 1 Comments

Point 1: The paper don’t bring great novelty (as new discussion or sorption mechanism, new material or good sorption capacity).

Response 1: Line 50-53, “we propose to modify calcined magnesite using ILs for CO2 capture. This approach can retain the benefits and solve the weakness of magnesite and ILs, respectively, designing an effective, cost efficiency, easily achieved and low energy consumption adsorbent.”

Solid adsorbent is more convenient than liquid adsorbent for industrial application, Line 30-32. Pure IL also has high-cost and high viscosity causing strong gas diffusion resistances. Among solid adsorbent, although the adsorption capacity isn’t too high, which is in medium level by comparing with other adsorbent, the magnesite is abundant and non-pollution nature resource and can produce MgO-based adsorbent simply, reducing much cost.  By introducing IL, the regeneration temperature is much lower than other MgO-based solid adsorbent. By experiments, the CO2 capacity is 0.81 mmol/g after eight cycles, indicating that the material has recyclability but need to improve. All of them are the benefits of our materials investigated in our paper.

Point 2:The use of the material for CO2 capture is not really viable, since it uses lot of energy for calcination and don’t show good recyclability.

Response 2: Among solid adsorbents, carbon-based, MgO-based, CaO-based, porous materials adsorbent have better CO2 adsorption performance. All of them need activation or regeneration under high temperature once or more, and the production also need cost. The preparation of MOFs also need energy and chemical agents. Magnesite only need high temperature once during the service life and the raw material is nature resource and don’t need deep processing. The CO2 capacity is 0.81 mmol/g after eight cycles, indicating that the material has recyclability but need to be improved. The comparison with cabon-based and MOFs adsorbent are in Line 252-258. The calcined condition of  MgO-based adsorbent also have been compared in Ref. 22, Table 2. By comparison, we consider that the calcination under high temperature only once can be accepted.  

Point 3:The increase in the amount of IL generates an increase in the sorption capacity, which may indicate that the ILs pure or mixed between IL and any residue (not calcined) are capable of capturing more CO2 than the material.

Response 3: Line 30-32, “Generally, liquid adsorbent of CO2 adsorption capacity is higher than solid adsorbent. However, high investment, corrosion of the equipment and high regeneration energy consumption are the problem for application in industry.” The drawbacks of pure ILs also make it difficult to apply in industry. Line 43-47, “Ionic liquids (ILs) can address such a problem and exhibit superior CO2 adsorption performance[24–25]; however, besides high cost, another drawback of pure ILs is their high viscosity causing strong gas diffusion resistances[26]. Therefore, for practical application, it will also be highly desirable to dispersedly immobilize ILs into a support in order to overcome the gas diffusion limitation. Therefore, we propose to modify calcined magnesite using ILs for CO2 capture. This approach can retain the benefits and solve the weakness of magnesite and ILs, respectively, designing an effective, cost efficiency, easily achieved and low energy consumption adsorbent”. Furthermore, the process equipment of liquid adsorbent also is different from the solid adsorbent and needs bigger investment.

The magnesite can be calcined to MgO, which also can contribute to CO2 capture and reported in Ref.22. By comparing with other materials shown in Table 2, the CO2 adsorption capacity of [APMIM]Br-modified magnesite is in medium level. However, the magnesite is abundant and non-pollution nature resource, saving a lot cost.

Point 4:The calcination part to produce the material consumes a lot of energy, and the authors have not demonstrated the advantage of this.

Response 4: Among solid adsorbents, carbon-based, MgO-based, CaO-based, porous materials adsorbent have better CO2 adsorption performance. All of them need activation or regeneration under high temperature once or more, and the production also need cost. The preparation of MOFs also need energy and chemical agents. Magnesite only need high temperature once during the service life and the raw material is nature resource and don’t need deep processing. By comparison, we consider that the calcination under high temperature only once can be accepted.

Point 5:Blank experiments are missing.

Response 5: The CO2 adsorption performance of pure magnesite has been added in Figure 5. The pure ILs and support ILs have their own benefits and drawbacks. Liquid adsorbent has the dominant benefit of high CO2 adsorption capacity, but the high cost and high viscosity causing strong gas diffusion resistances of pure ILs also make it difficult to apply in industry, Line 43-47. Generally, liquid adsorbent of CO2 adsorption capacity is higher than solid adsorbent and the process equipment also have difference. We consider that it is inappropriate to compare support ILs with pure ILs only from the aspect of the CO2 adsorption capacity.  

Point 6:"torus imidazole" is an unusual expression;

Response 6: all "torus imidazole" in the manuscript have been corrected to “imidazole ring”.

Point 7:How can the relative pressure be 0 MPa?

Response 7: In experiments,  CO2 pressure was adjusted using a counterbalance valve. The flow rate in our experiments is small. When the counterbalance value open fully and the flow rate is small,  all the gas drains and can’t form resistance or pressure, so it display “0”.

Point 8:All graphs (adsorption curves) are similar, and only show the adsorption rate, which does not represent the most relevant information for the article. It should be more interesting to move part of the graph to supporting information and create figures/graphs comparing the sorption capacity of each material.

Response 8: CO2 adsortpion capacity was calculated by integration of adsroption curves. In order to  reflect the intuitive information directly, the respective values of CO2 adsorption capacity have been add in Figs 4-8 and the equation for calculating CO2 adsorption capacity has been added in Line 102-105.

Point 9:I am still not convinced with the kinetic experiment, the fit is similar in many models, this part should be further explored to prove its conclusion.

Response 9: According to other reviewer’s suggestion, paragraph “3.9. Adsorption Kinetics” has been removed. I think it’s better to put this part together with the adsorption thermodynamics in subsequent work.

Reviewer 2 Report

The Authors resubmitted the manuscript after a careful revision. All my comments to the original version of the manuscript have been addressed and the manuscript has been improved accordingly. Therefore, my suggestion is to accept it for publication.

Author Response

Thank reviewer recognition and suggestions.

Reviewer 3 Report

The paper presents research on CO2 adsorption and kinetics of ionic liquid-modified calcined magnesite. The presentation of methods and scientific results in the current form is satisfactory for publication in the Nanomaterials journal. The minor and significant drawbacks to be addressed can be specified as follows:
1.    Fig. 2. What is the cause of the peak between 1000 and 1169 cm-1 (visible in (3) and (4), especially)?
2.    Fig. 3(a). It should have the same resolution as the others, i.e 1 micrometre. See also (e) and (f). 
3.    Fig. 4, see also others (Figs. 5-8), legend. Would you please add the respective values of CO2 adsorption capacity (taken from Tab. 1). In this form, the plots are not entirely interesting and legible. This form of data presentation (c/c0) makes it difficult to interpret the results.
4.    Figs. 5-8. Are all the given data descriptive of equations? What to do with it?
5.    Figs 7 and 8 should be combined.
6.    Fig. 9, figure captions. Please add [APMIM]Br.
7.    Fig. 10, y-axis. mW mg-1 ---> mW/mg.
8.    Fig. 10. In my opinion, the data for pure magnesite should also be published in this figure.
9.    Fig. 12. The adsorption isotherms are very strange. Very large area of linearity of the Henry isotherm. In my opinion, the data for pure magnesite should also be published in this figure.
10.    Paragraph “3.9. Adsorption Kinetics” does nothing to work - I suggest removing it.
11.    Fig. 13. The points should be added instead of solid line (experimental data).
12.    The conclusions are bland.

Author Response

Response to Reviewer 3 Comments

Point 1: Fig. 2. What is the cause of the peak between 1000 and 1169 cm-1 (visible in (3) and (4), especially)?

Response 1: Line131-132: The peaks of magnesite modified using [AEMIM]BF4 and [APMIM]BF4 at 1083 cm1 and 1126 cm1 is caused by B-F bond.

Point 2:Fig. 3(a). It should have the same resolution as the others, i.e 1 micrometre. See also (e) and (f). 

Response 2: The resolution of all pictures in Figure 3 has been checked.

Point 3: Fig. 4, see also others (Figs. 5-8), legend. Would you please add the respective values of CO2 adsorption capacity (taken from Tab. 1). In this form, the plots are not entirely interesting and legible. This form of data presentation (c/c0) makes it difficult to interpret the results.

Response 3: The respective values of CO2 adsorption capacity have been add in Figs 4-8.

Point 4: Figs. 5-8. Are all the given data descriptive of equations? What to do with it?

Response 4: Line 102-105: The equation for calculating CO2 adsorption capacity has been added.

The CO2 adsorption capacity was calculated by following equation:

CO2 capacity = mmol/g

where A is the integral area by adsorption curves in min, S is the CO2 concentration, M is the CO2 flow rate in mL / min, C is constant 22.4 L/mol, m is the quality of adsorbent in g.

Point 5: Figs 7 and 8 should be combined.

Response 5: Line 227:  Figure 8 was placed Figure 7 by mistake. It has been corrected.

Point 6:Fig. 9, figure captions. Please add [APMIM]Br.

Response 6: [APMIM]Br has been added in Figure 9 caption.

Point 7: Fig. 10, y-axis. mW mg-1 ---> mW/mg.

Response 7: mW mg-1 has been corrected to  mW/mg in Figure 10

Point 8:Fig. 10. In my opinion, the data for pure magnesite should also be published in this figure.

Response 8: The TG-DSC date has been published in our literature on the investigation of the magnesite calcined condition. Line 276-277: “The TG-DSC curve of pure magnesite has been reported in our previous work[22], showing that the decomposition of magnesite to MgO happenes from 400 oC to 800 oC” is added. 

Point 9: Fig. 12. The adsorption isotherms are very strange. Very large area of linearity of the Henry isotherm. In my opinion, the data for pure magnesite should also be published in this figure.

Response 9: The BET has been retested to check the adsorption isotherms under the same condition with the same apparatus and nearly the same results was achieved. The BET data for pure magnesite has published in our previous work, Ref.22, but the comparison has been added in Line 309-311 “ By compared with the BET surface area of the pure calcined magnesite 118.6 m2/g[22], it indicats that 30% IL loading amount has little influence on pore structure of magnesite modified by [APMIM]Br”.

Point 10: Paragraph “3.9. Adsorption Kinetics” does nothing to work - I suggest removing it.

Response 10: Paragraph “3.9. Adsorption Kinetics” has been removed.

Point 11:Fig. 13. The points should be added instead of solid line (experimental data).
Response 11: Figure 13 in Paragraph “3.9. Adsorption Kinetics” has been removed.

Point 12: The conclusions are bland.

Response 12: The conclusions have been rewritten.

Round 2

Reviewer 1 Report

Despite the great effort of the author to improve the article, the work still does not present great news (new discussion, sorption mechanism, new material or good sorption capacity) for Nanomaterials journal. Furthermore, the kinetic part, which could represent an important discussion of the work, was removed from the paper rather than being further explored. This fact does not represent a good attitude to be taken.

For these reasons, I don’t recommend this paper for publication.

Author Response

Point 1: Despite the great effort of the author to improve the article, the work still does not present great news (new discussion, sorption mechanism, new material or good sorption capacity) for Nanomaterials journal.

Response 1: Magnesite is a rich mineral resource in China. A lot of magnesite production processing belong to primary products and resource consumption. The MgO based adsorbent can be easily achieved by calcining abundant, low cost and non-toxic magnesite. For the CO2 adsorbents, except for the capture capacity, another important issue is the cost. We explored a new material, modify calcined magnesite using ILs for CO2 capture. This approach can retain the benefits and solve the weakness of magnesite and ILs, respectively, designing an effective, cost efficiency, easily achieved and low energy consumption adsorbent.

Point 2:Furthermore, the kinetic part, which could represent an important discussion of the work, was removed from the paper rather than being further explored. This fact does not represent a good attitude to be taken. I am still not convinced with the kinetic experiment, the fit is similar in many models, this part should be further explored to prove its conclusion.

Response 2: We are very sorry for removing the kinetic part in accordance with the other reviewer’s comments without clearly explanation to you. This part was not intentionally removed. We do the research carefully and do best efforts to revise the article according to your suggestion. The kinetic part has been recovered. In experiments, we investigated the CO2 adsorption performance for 60 min. Therefore, in kinetics part, the time was extended to 60 min. The R2 values of pseudo-first-order and Bangham models are still 0.999. The parameter n of Bangham is also 1, indicating that the Bangham model is completely the same as the pseudo-first-order model. The R2 value of pseudo-second-order model is lower, 0.988. The tendency of pseudo-second-order curve also don’t fit the experimental curve.

Reviewer 3 Report

The authors have made a substantial improvement for this article. The manuscript can be accepted for publishment in the present form.

Author Response

Thank reviewer‘s recognition and suggestions.

Round 3

Reviewer 1 Report

-

This manuscript is a resubmission of an earlier submission. The following is a list of the peer review reports and author responses from that submission.

Round 1

Reviewer 1 Report

Wang et. al. prepared adsorbent materials from ionic liquids (ILs) and calcined magnesites to capture CO2. They studied different porecetage of IL to create the material and evaluate different CO2 capture parameters. However, the work does not bring significant news. The sorption capacity is low (1.34 mmol / g), lower than many other examples in the literature. Furthermore, the material does not represent a scientific advance. The article should improve the scientific language, English and deepen the discussion. I do not suggest this article for publication.

Some suggests and comments:
- Page 1: When is written  "academics at home and abroad", did you mean research groups?
- There are many terms repeated throughout the document, for example, page 1 and 2: adsorbents.
- Page 2, line 44-48: the language used is colloquial.
- The abbreviation IL is unusual, use a more accepted abbreviation (ie aemim, NH3emim).
- Why do you call the material a nanomaterial? Did you perform any characterization to confirm this?

Experimental part
Use scientific terms and correct English. Remove some inappropriate terms as 
"ingredient".
- Page 2, line 76-78. Were two calcinations carried out?

Results:
- Figure 2-6: Clarify for the reader what is "c c0" and what do you see in the graph. All adsorption curves are very similar and do not show any significant information. The discussion around adsorption capacity does not fit the adsorption curves. Adsorption kinetics and time to stability should be further explored in the graph and were usually not mentioned.
- Page 5, line 142: Increasing the amount of IL, genereates a increase the sorption capacity. Did you run the experiments in blanck, only with IL and only with MgO? Many articles demonstrate good CO2 capture value for amine ILs. Is it possible that the use of MgO decreases IL capacity? Tests with IL neat should show up on paper.
- Page 5 line 156. Why does 50ºC have better sorption capacity than 30ºC? How could you explain this unusual behavior?
- Page 7, line 182 and figure 6. What do you mean by 0 MPa? Atmospheric pressure? 
- Figure 7. Recycling: The graph showed that the material is not suitable for recycling. How do you explain this? Is is possible any IL leaching to occur, causing a material modification? 90ºC still high temperature and consuming energy. There are already many reports on the application of N2 at room temperature.

- The characterization of the material must appear before the application.
- Page 9, line 213. What do you mean temperature improvement? Should it be increased?
Is mass loss attributed to micromolecules or impurities, what kind of impurities should be present?
- Page 10: What do you mean by "torus imidazole strength"? It's an unusual expression.
- The characterization is not well explored. The discussion about this is deficient and vague.
- Figure 11 and kinetic results: The pseudo first order and Bangham have the same results. What could you conclude with kinetic experiments? Was the Freudlich and Langmuir adsorption model tested? Fig 13 shows really similar adjustments, the study should be properly concluded, seems incomplete this part of the paper. 

Conclusion
The author reported that they demonstrated a sorbent with excellent CO2 capture capacity with low energy cost of regeneration. A comparison with the literature is lacking, with a proper review of the state of the art, these conclusions are not entirely true. Some pure ILs are capable of capturing more CO2 than the material; the calcination part to produce the material consumes lot of energy, and the authors have not demonstrated the advantage of this.

Reviewer 2 Report

Review report attached as pdf file.

Reviewer 3 Report

Please see the attached file with the review comments.
